# Iron Limitation Restores Autophagy and Increases Lifespan in the Yeast Model of Niemann–Pick Type C1

**DOI:** 10.3390/ijms24076221

**Published:** 2023-03-25

**Authors:** Telma S. Martins, Rafaela S. Costa, Rita Vilaça, Carolina Lemos, Vitor Teixeira, Clara Pereira, Vítor Costa

**Affiliations:** 1i3S—Instituto de Investigação e Inovação em Saúde, Universidade do Porto, 4200-135 Porto, Portugal; 2IBMC—Instituto de Biologia Molecular e Celular, Universidade do Porto, 4200-135 Porto, Portugal; 3Instituto de Ciências Biomédicas Abel Salazar, Universidade do Porto, 4050-313 Porto, Portugal

**Keywords:** Ncr1, TORC1, Aft1, vacuoles, vesicle trafficking, iron, autophagy, oxidative stress, chronological aging

## Abstract

Niemann–Pick type C1 (NPC1) is an endolysosomal transmembrane protein involved in the export of cholesterol and sphingolipids to other cellular compartments such as the endoplasmic reticulum and plasma membrane. NPC1 loss of function is the major cause of NPC disease, a rare lysosomal storage disorder characterized by an abnormal accumulation of lipids in the late endosomal/lysosomal network, mitochondrial dysfunction, and impaired autophagy. NPC phenotypes are conserved in yeast lacking Ncr1, an orthologue of human NPC1, leading to premature aging. Herein, we performed a phosphoproteomic analysis to investigate the effect of Ncr1 loss on cellular functions mediated by the yeast lysosome-like vacuoles. Our results revealed changes in vacuolar membrane proteins that are associated mostly with vesicle biology (fusion, transport, organization), autophagy, and ion homeostasis, including iron, manganese, and calcium. Consistently, the cytoplasm to vacuole targeting (Cvt) pathway was increased in *ncr1*∆ cells and autophagy was compromised despite TORC1 inhibition. Moreover, *ncr1*∆ cells exhibited iron overload mediated by the low-iron sensing transcription factor Aft1. Iron deprivation restored the autophagic flux of *ncr1*∆ cells and increased its chronological lifespan and oxidative stress resistance. These results implicate iron overload on autophagy impairment, oxidative stress sensitivity, and cell death in the yeast model of NPC1.

## 1. Introduction

Lysosomes and yeast lysosome-like vacuoles are acidic organelles that play critical roles in the degradation of material delivered by autophagy or by the endocytic pathway, compartmentation of metabolites, including ions and amino acids, and nutrient signaling. Plus, they interact with other organelles, including mitochondria and endoplasmic reticulum (ER), to promote cellular homeostasis [1,2,3]. The functional decline of lysosomes has been implicated in aging and numerous diseases. For example, lysosomal acidification and autophagy decrease with age, leading to defects in nutrient storage and in the degradation of damaged proteins and organelles, cellular dyshomeostasis, and ultimately to cell death [4]. Lysosomal impairment and defects in the lysosomal-mitochondrial axis are also hallmarks of lysosomal storage disorders (LSDs), including Niemann–Pick type C (NPC) disease [5].

NPC disease is a rare LSD caused by loss-of-function mutations in NPC1, in 95% of cases, or NPC2 [6,7]. It affects nearly 1/120,000 live births with clinical manifestations such as malfunction of the liver and lung and progressive neurodegeneration that cause ataxia and neurological dysfunction [8,9,10]. NPC1 and NPC2 are late endosomal/lysosomal proteins involved in the transport of endocytosed cholesterol. NPC2 is a soluble glycoprotein that has a high affinity to cholesterol and transfers it to NPC1, a transmembrane protein that promotes cholesterol export and transfer to the ER, plasma membrane, recycling endosomes, trans-Golgi network, and mitochondria [11,12,13]. In NPC disease, unesterified cholesterol accumulates in endolysosomal compartments, concomitantly with sphingolipids (e.g., sphingomyelin, glycosphingolipids, and sphingosine), which become depleted in other cellular compartments [14,15].

Disruption of cholesterol dynamics in NPC results in vesicular trafficking defects, a decrease in endosome and lysosome motility, and stalled autophagic flux associated with induction of autophagy and defects in the fusion of autophagosomes with late endosomes/lysosomes [9,16,17,18,19,20,21,22,23,24,25,26]. A proteomic analysis of isolated lysosomes showed that loss of NPC1 leads to the upregulation of several proteins associated with autophagy, namely autophagosome elongation and closure (GABARAPL2), endocytosis (RAB5A), lysosomal biogenesis (CLN3, IGF2R), and selective autophagy (SQSTM1/p62) [27]. Different cathepsins are also upregulated in NPC [28,29], but the activity of cathepsin D decreases due to defects in its maturation [30]. Several pieces of evidence indicate that lysosomal hydrolases are depleted and autophagy is inhibited in cell models of NPC1 disease due to an aberrant activation of the mammalian Target of Rapamycin Complex 1 (mTORC1) by cholesterol [31,32,33]. Conversely, cholesterol depletion with methyl-β-cyclodextrin restores the autophagic flux in NPC1 cells by inducing AMP-activated protein kinase (AMPK)-dependent activation of ULK1/ATG1 and inhibition of mTORC1 [34]. Nevertheless, other studies reported that mTORC1 is inhibited [35] or not affected [16] in NPC disease.

Defects in autophagy and mitophagy, concomitant with a decrease in mitochondrial motility, lead to mitochondrial impairment in NPC disease [18]. Indeed, NPC1 fibroblasts exhibit depletion of mtDNA and changes in mitochondrial membrane potential and mitochondrial dynamics [36]. However, several other factors have been implicated in mitochondrial dysfunction. These include the downregulation of sphingosine-1-phosphate receptors and sphingosine-1-phosphate signaling that leads to KLF2- and ETV1-dependent transcriptional repression of mitochondrial biogenesis [37,38]. Notably, mitochondrial dysfunction of NPC1 cells is also caused by the accumulation of cholesterol due to the expansion of lysosome-mitochondria contact sites, which increases cholesterol transfer to mitochondria, disrupting its function and redox homeostasis [39,40]. Mitochondrial dysfunction in NPC has also been associated with calcium cytotoxicity resulting from the activation of the IP3R type 1 in the ER that increases calcium release [41]. Numerous studies have also shown that iron homeostasis is disrupted in the plasma and tissues of NPC patients and in animal and cell models of this disease [42,43,44,45,46,47]. Iron is a redox-active metal that catalyzes the Fenton reaction, in which hydrogen peroxide is converted into highly reactive hydroxyl radicals. Thus, iron accumulation has been linked to neurodegenerative disorders [48] and may contribute to oxidative distress and ferroptosis in NPC [49].

The yeast *Saccharomyces cerevisiae* has been used as a model for neurodegenerative diseases [50] and LSDs, including NPC [51]. Yeast cells express orthologues of NPC1 (named Ncr1) and NPC2, which are localized to lysosome-like vacuoles and involved in lipid trafficking [52,53]. Recent studies showed that sterols are delivered to the luminal membrane leaflet by the N-terminal domain of Ncr1 and transported by a proton-driven mechanism [54]. Ncr1 and Npc2 play a role in the formation of sterol-rich membrane domains that are required for microlipophagy [55]. Notably, yeast cells lacking Ncr1 recapitulate molecular features of NPC including trafficking defects [52], impaired transport of sterols to the vacuolar membrane under starvation conditions [54], accumulation of long-chain sphingoid bases and ceramide leading to a shortened lifespan [56,57], mitochondrial dysfunction [36,56,57], and changes in calcium and metal ion homeostasis and lipid trafficking [36], as well as alterations in sphingolipid signaling and nutrient signaling [36,56,57] and in cytoskeleton organization [36].

In this work, we performed an MS-based phosphoproteomic analysis of the vacuolar membranes from *ncr1*∆ mutant cells to characterize how the loss of Ncr1 affects yeast lysosome-like vacuoles. Proteins associated with vesicle fusion, transport, and organization and with iron homeostasis were the most significantly altered. We found that the cytoplasm to vacuole targeting (Cvt) pathway was induced and autophagy was impaired in *ncr1*∆ cells. Notably, *ncr1*∆ cells exhibited iron overload, and iron deprivation restored the autophagic flux and improved the longevity and oxidative stress resistance of *ncr1*∆ cells. Overall, these results implicate iron overload on autophagy impairment and cell death in the yeast model of NPC1.

## 2. Results and Discussion

### 2.1. Characterization of the Vacuolar Membrane Proteome in the Yeast Model of NPC1

To characterize how Ncr1 affects cellular functions mediated by the yeast lysosome-like vacuoles, the effects of Ncr1 deficiency on the phosphoproteome of vacuolar membranes were investigated. For that, vacuolar membranes were isolated from wild-type and *ncr1*∆ cells, and samples were analyzed by nanoLC-MS/MS (an overview of the experimental design is shown in Figure 1). Vacuolar membranes were analyzed instead of intact vacuoles to avoid the interference of non-vacuole constituents that are delivered from other compartments for degradation. The enrichment and purity of the vacuolar membranes were qualitatively analyzed by Western blot using antibodies against specific organelle markers (not shown).

We were able to detect 230 proteins annotated as vacuolar membrane proteins, representing around 80% of the total number of vacuolar membrane proteins in *S. cerevisiae* (protein levels and statistical analysis are presented in Appendix A). From these, we identified phosphorylated residues in 54 proteins (Appendix A contains information about the percentage of phosphorylation of each identified residue and the respective statistical analysis). Our results showed that the levels of 45 proteins changed significantly in *ncr1*∆ vacuolar membranes, and 18 proteins were differently phosphorylated, with 4 proteins (Yck3, Ybt1, Fet5, and Fth1) presenting alterations in both levels and phosphorylation (Figure 2A–C). In *ncr1*∆ vacuolar membranes, 21 proteins were less abundant and 24 proteins more abundant. Moreover, 14 amino acid residues were less phosphorylated and 8 more phosphorylated. Gene ontology (GO) analysis on biological processes showed an enrichment in proteins related to vesicle and vacuole fusion and organization, ion homeostasis and transport (namely, iron, manganese, and calcium), polyphosphate metabolism, Golgi organization, autophagy, and protein targeting to vacuole (Figure 2D and Appendix A). These results suggest that vesicle trafficking pathways and ion homeostasis are altered in yeast cells lacking Ncr1, being consistent with previously reported trafficking defects and calcium/iron dyshomeostasis [9,16,17,18,19,20,21,22,23,24,25,26,42,43,44,46,47,58] and alterations in the expression of lysosomal proteins related to autophagy in NPC [27]. A previous study with the yeast model also showed that Ncr1 interacts with vacuolar proteins, including the Pmc1 Ca^2+^ ATPase and the Fth1 iron transporter. Moreover, the yeast model of NPC exhibits mislocalization of proteins involved in vacuole protein sorting (Vps41, Did4, Pmr1, and Yip5), and *NCR1* is synthetically lethal with genes associated with protein sorting (*MVB12* and *APS1*) [36].

### 2.2. Cvt Pathway Is Induced and Autophagy Compromised in the Yeast Model of NPC1

Trafficking pathways are critical for vacuole biogenesis and function. In eukaryotic cells, organellar proteins are synthesized in the ER and then transferred to the Golgi apparatus for sorting into their cellular compartments. In yeast, proteins are targeted to the vacuole either through endosomes via the vacuolar protein sorting (VPS/CPY) pathway or directly through the alkaline phosphatase (AP-3) pathway [59]. Our proteomic data revealed that *ncr1*∆ vacuolar membranes had lower levels of several proteins (Yck3, Vam3, Nyv1) transported to the vacuole through the AP-3 pathway, suggesting a defective route. To assess the AP-3 pathway, cells were transformed with a plasmid expressing a GFP-tagged Nyv1-Snc1 fusion protein that localizes to both vacuolar and plasma membrane in cells defective in AP3 transport (e.g., *yck3*∆ cells) [60] (Figure 3A). Our results showed that GFP-Nyv1-Snc1 was restricted to the vacuolar membrane in wild-type and *ncr1*∆ cells, suggesting that the AP-3 pathway is not compromised (Figure 3A). We also observed that Yck3-S48 phosphorylation decreased in the *ncr1*∆ mutant (Appendix A). However, this alteration does not seem to affect the AP-3 pathway and, therefore, may not impact *ncr1*∆ phenotypes.

Carboxypeptidase Y (CPY) is the best-studied soluble protein targeted to the vacuole through the CPY pathway and has been largely used as a read-out of this pathway [61]. CPY is transported to the vacuole where it is processed to its mature form by Pep4-dependent proteolytic activation [62]. To assess this pathway, CPY was analyzed by Western blotting. Cells lacking Pep4 and wild-type cells treated with rapamycin were also analyzed to facilitate the identification of the precursor and mature forms of CPY. The results revealed that CPY was fully processed in both wild-type and *ncr1*∆ cells, indicating that the CPY pathway is not defective in Ncr1-deficient cells (Figure 3B). These results are consistent with a previous study showing that Ncr1 deficiency does not affect endocytosis, CPY maturation, or the AP-3 pathway [63].

The cytoplasm to vacuole targeting (Cvt) pathway is another biosynthetic route for vacuolar proteins in yeast [64]. This pathway uses most of the molecular machinery of autophagy to transport vacuolar hydrolases such as aminopeptidase I (Ape1) and α-mannosidase (Ams1). Biosynthetic cargo is sequestrated by a cytosolic double-membrane vesicle that subsequently fuses with the vacuole, allowing the resident enzymes access to their final destination. Our proteomic analysis showed a 13.6-fold increase in Ams1 in the vacuolar membranes of the *ncr1*∆ mutant. This led us to postulate that the Cvt pathway may be induced in this mutant. To test this hypothesis, we analyzed the processing of ApeI to its mature form by Western blotting, a well-established assay for Cvt. Since this pathway is induced during growth from exponential (fermentative) to post-diauxic shift (PDS; respiratory) phase, we assessed Cvt in both phases. The results demonstrated that the levels of the mature form of ApeI increased in *ncr1*∆ cells grown to the exponential phase (Figure 3C), suggesting that the Cvt pathway flux is enhanced in cells lacking Ncr1. It should be noted that ApeI was not detected in our study since it is not a vacuolar membrane protein.

In addition to the Cvt pathway, autophagy is induced during growth to the PDS/respiratory and stationary phases. These cellular adaptations are important for cell longevity and their impairment has been associated with numerous diseases, including neurodegenerative disorders and aging [4]. We assessed the autophagic flux by monitoring the processing of GFP-Atg8 [65]. Since the GFP moiety is resistant to proteolysis and remains stable upon vacuolar delivery of GFP-Atg8, the detection of a free GFP signal by Western blotting can be used to evaluate the autophagy flux. Autophagy is a crucial process for cell survival that is negatively regulated by TORC1 [66]. Autophagy is induced as cells start to age and nutrients become scarce, removing damaged macromolecules that are recycled and allowing cells to survive under nutrient deprivation. In agreement, we observed an increase in autophagy in wild-type cells grown to the PDS phase or treated with rapamycin (to inhibit TORC1). Contrary to the wild type, cells lacking Ncr1 were unable to induce autophagy at the PDS phase (Figure 4A), which may underlie the premature aging observed in these cells [56]. Changes in autophagy have been largely described in NPC1-deficient mammalian cells. The number of autophagosomes increases but autophagic progression and proteolysis are stalled [16,20,23,67]. This has been mostly associated with defects in the cargo-targeting step and in the fusion step of autophagosomes and lysosomes to form autolysosomes. The role of TORC1 signaling in the autophagy defects of NPC1-deficient cells is controversial. Some studies report that the loss of function of NPC1 results in TORC1 hyperactivation and that inhibition of TORC1 signaling is sufficient to restore lysosomal function and autophagy [31,32,33]. However, others reported no alterations or even a decrease in TORC1 activity upon NPC1 deficiency [20,35]. To evaluate TORC1 activity, we assessed TORC1-mediated phosphorylation of ribosomal protein S6 (Rps6 in yeast) at S232/233 (S235/236 in human S6), a specific readout for TORC1 activity [68]. TORC1 responds to growth conditions and nutrient availability and, as expected, its activity decreased in wild-type cells grown to the PDS phase, which agrees with the induction of autophagy. Our results also demonstrated that cells lacking Ncr1, compared to wild-type cells, displayed significantly lower levels of phospho-Rps6 (Figure 4B,C). The phosphorylation of Sch9, another target of TORC1, also decreased in *ncr1*∆ cells (Appendix A). Overall, these results indicate that TORC1 signaling is downregulated in the yeast model of NPC1, suggesting that the autophagic flux is impaired in a TORC1-independent manner.

### 2.3. Ncr1 Deficiency Induces the Iron Regulon and Aft1-Dependent Iron Overload

Iron is an important cofactor essential for cell viability, but iron in excess increases the formation of harmful hydroxyl radicals through the Fenton reaction, promoting oxidative stress [69]. Loss of iron homeostasis associated with NPC1 disease has been documented in several studies. Hung et al. reported higher levels of iron in the brain and decreased levels in the liver and spleen of the NPC1 mouse model. A tendency for iron levels to be higher in post-mortem human NPC1 cerebellar tissue and lower in plasma was also observed [42]. Recently, Chen et al. detected lower levels of ferritin in the cerebrospinal fluid of NPC1 patients, consistent with alterations in iron homeostasis in the central nervous system [46]. Indeed, the authors showed that Npc1^−/−^ mice and NPC1 patients have significantly reduced serum iron and several hematological changes characteristic of iron deficiency anemia. A lower expression of L-ferritin and reduced iron levels in the liver of Npc1^−/−^ mice was also observed together with higher levels of serum ferritin in NPC patients [46]. NPC1 fibroblasts from patients also exhibit enhanced expression of genes encoding proteins involved in iron homeostasis (ferritin and sideroflexin 1) [70]. However, several studies report a ferritin deficiency in various NPC tissues [44,46,71,72] and a decrease in FTH1 (H-ferritin) protein levels in NPC1-deficient HeLa cells [47]. It has been proposed that lower levels of ferritin lead to an excess of unbound iron, thus favoring lipid peroxidation caused by reactive oxygen species (ROS) [73]. Recently, it was demonstrated that changes in autophagy contribute to the loss of iron homeostasis upon NPC1 deficiency [47]. Abnormal autophagy in NPC1-deficient HEI-OC1 auditory cells induces ferritinophagy (autophagy-dependent iron release from ferritin), resulting in loss of iron homeostasis and increasing lipid peroxide that induces ferroptosis [47].

Interestingly, our proteomic analysis showed that around 24% of proteins with increased levels at vacuolar membranes of cells lacking Ncr1 were associated with iron homeostasis (Fth1, Fet5, Smf3, Fre8, and Enb1). We also identified alterations in the phosphorylation status of three iron homeostasis-related proteins: Fth1, Fet5, and Ccc1 (Appendix A). This led us to assess whether iron levels were altered in *ncr1*∆ cells. Although iron content was similar in wild-type and *ncr1*∆ cells grown to the exponential phase, we observed that *ncr1*∆ cells accumulate iron at the PDS phase (Figure 5A). This led us to postulate that loss of iron homeostasis may be a contributing factor for oxidative stress sensitivity and premature aging of *ncr1*∆ cells [56]. Changes in iron levels cannot be attributed to growth defects of the *ncr1*∆ mutant as they were analyzed at time points where both strains clearly entered the PDS phase (see Appendix A). Since proteins involved in vacuolar iron mobilization were upregulated and vacuolar iron storage is known to protect against iron toxicity [74], iron is probably accumulating in the cytoplasm and, as consequence, it may also accumulate in other organelles. Further research is needed to characterize how the dysregulation of vacuolar iron transporters, including changes in its phosphorylation, impacts *ncr1*∆ phenotypes.

Iron uptake and mobilization in the yeast *S. cerevisiae* is mainly regulated at the transcriptional level by the Aft1 transcription factor. When iron availability is low, Aft1 upregulates the iron regulon, a set of genes associated with iron import and mobilization of vacuolar iron stores. However, activation of Aft1 in iron-replete conditions can result in iron overload [75]. To assess the involvement of Aft1 in the accumulation of iron in the *ncr1*∆ mutant, cells were transformed with a plasmid containing the Aft1 binding sequence from *CTH2* promoter fused to a LacZ reporter, and β-galactosidase activity was determined. As expected, *CTH2*-LacZ was induced in wild-type cells treated with the iron chelator bathophenanthrolinedisulfonic acid (BPS) (Figure 5B). Under normal conditions, β-galactosidase activity was significantly higher in *ncr1*Δ cells than in parental cells. Moreover, iron overload was abolished in *ncr1*Δ*aft1*Δ cells (Figure 5C), indicating that early activation of Aft1 (in the exponential fermentative phase) mediates iron accumulation in a later stage (PDS phase) in *ncr1*∆ cells. The vacuolar defects described in this study for *ncr1*∆ cells might contribute to the iron overload in response to abnormal activation of Aft1 (lack of vacuolar iron buffering capacity). On the other hand, premature aging of *ncr1*∆ cells can lead to defective intracellular iron homeostasis and abnormal activation of the iron regulon (Aft1) [76]. *AFT1* deletion did not suppress the premature aging of cells lacking Ncr1 (Appendix A). However, *aft1*∆ cells also exhibited a shortened lifespan. Since Aft1 also has iron-independent functions [77], *AFT1* deletion likely affects other processes important for cell longevity, which may explain why it did not increase the lifespan of *ncr1*Δ cells. Moreover, *aft1*∆ cells display mitochondrial defects similar to the *ncr1*∆ mutant, which can also explain its shortened lifespan [76]. The mechanism leading to Aft1 activation in cells lacking Ncr1 remains to be disclosed. These cells exhibit glutathione depletion [56], which may contribute to Aft1 activation as dysregulation of glutathione levels triggers a starvation-like response [78]. Depletion of glutathione and mitochondrial dysfunction in *ncr1*∆ cells might impair ISC assembly machinery and compromise the iron-dependent nuclear export of Aft1 [79]. Changes in lipids may also contribute to iron dyshomeostasis, as previous studies have unraveled a functional crosstalk between lipid metabolism and iron maintenance [74]. Ergosterol [80] and sphingolipids [81], both altered in *ncr1*∆ cells [56,57,82], modulate Aft1 activity upon iron depletion. This raises the hypothesis that the accumulation of ergosterol and sphingolipids in *ncr1*∆ cells could contribute to Aft1 deregulation.

### 2.4. Iron Chelation with BPS Increases Lifespan, Oxidative Stress Resistance, and Restores Autophagic Flux in the Yeast Model of NPC1

Recently, it was shown that iron limitation, by treating cells with BPS, prolongs yeast lifespan in an autophagy-dependent manner [83]. Since cells lacking Ncr1 display an impairment in autophagic flux and iron accumulation, this led us to test whether BPS would produce beneficial effects. For that, we analyzed the chronological lifespan. We considered the PDS phase as t0 in this assay, instead of the beginning of the stationary phase (3 days after PDS; standard conditions used in most studies), as *ncr1*∆ cells have a short lifespan [56,57]. Our results showed that BPS treatment caused a significant lifespan extension in *ncr1*∆ mutant cells (Figure 6A,B), indicating that iron accumulation is involved in the *ncr1*∆ mutant shortened lifespan. In the wild-type strain, however, BPS did not extend longevity, and even decreased it (Figure 6A,B), contrary to a previous report [83]. This may arise from differences in growth conditions since, in addition to BPS supplementation, Montella-Manuel et al. used an SD medium prepared with an iron-free nitrogen base [83]. Notably, incubation with BPS also suppressed the hydrogen peroxide sensitivity and high ROS levels of *ncr1*Δ cells (Figure 6C,D), corroborating the initial hypothesis that iron overload promotes intracellular oxidation and oxidative stress sensitivity in cells lacking Ncr1. Studies using a mouse model of NPC1 suggest that iron chelation with deferiprone (DFP) is ineffective in slowing the disease progression or in prolonging lifespan [43]. However, another study reported that DFP conjugated with β-cyclodextrin partially rescues the increase in lysosomal volume in NPC1-deficient cells [84]. Thus, more studies are required to evaluate the use of iron chelators as potential therapeutical strategies against NPC.

To further investigate if the beneficial effects imparted by iron chelation were associated with an improvement of other *ncr1*Δ phenotypes, we assessed mitochondrial respiration and autophagy in the mutant upon BPS treatment. Our results showed that BPS did not suppress the mitochondrial dysfunction exhibited by *ncr1*Δ cells, suggesting that it is not mainly caused by iron overload (Figure 6E). Interestingly, BPS significantly reduced the oxygen consumption rate in wild-type cells, which is consistent with defective respiration of cells in iron-depleted medium or *aft1*∆ mutant cells [85]. As mitochondrial function is crucial for lifespan [86], this result may explain why iron limitation decreased the chronological lifespan of wild-type cells (Figure 6A,B). To monitor autophagy, we used the GFP-Atg8 assay (Figure 6F–H). We observed that Atg8 levels tend to increase in wild-type cells treated with BPS, although the autophagic flux was not affected. Notably, BPS induced autophagy and restored the autophagic flux of cells lacking Ncr1, suggesting that iron chelation may improve *ncr1*Δ longevity by restoring the autophagic flux.

Several compounds are known to increase lifespan in part by decreasing free iron, protecting against oxidative damage [87]. Iron deficiency downregulates TORC1 activity [88], whose inhibition is largely associated with lifespan extension and healthspan in several organisms [89]. In yeast, iron limitation increases lifespan by inhibiting TORC1 and inducing autophagy [83]. Epigallocatechin gallate, an iron chelator, extends mice and rats’ lifespans by increasing the activation of autophagy and decreasing oxidative stress [90,91]. As TORC1 activity was lower in *ncr1*∆ cells (Figure 4B), the increase in the autophagic flux of the mutant treated with BPS is, probably, TORC1 independent. These data point to the existence of another mechanism by which autophagy is regulated by iron. Our results also implicate that loss of iron homeostasis impacts the autophagic flux in the yeast model of NPC disease. The identification of the molecular players involved in this regulation will contribute to our understanding of the molecular basis of the NPC1 disease. Notably, high levels of iron increase sphingolipid synthesis and sphingolipid signaling through the Pkh1 protein kinase [92], and we have previously shown that phytosphingosine (PHS) but not dihydrosphingosine (DHS) levels increase in *ncr1*∆ cells, leading to Pkh1 activation and a shortened chronological lifespan [56]. Plus, the hydroxylation of DHS to PHS is mediated by the iron-dependent protein Sur2 [93,94], and iron chelation decreases the accumulation of PHS, but not of DHS [95]. More studies are required to define if iron depletion increases the longevity of the yeast model of NPC1 through modulation of Pkh1 signaling. In summary, our results suggest that vesicle trafficking pathways and iron homeostasis are deregulated in cells lacking Ncr1 and implicate iron overload on the high oxidative stress sensitivity, impaired autophagic flux, and premature aging exhibited by the yeast model of NPC1 (Figure 7). This study supports a link between iron toxicity and NPC phenotypes, but more studies are required to clarify if iron chelation may be explored as a therapeutic strategy aimed at counteracting the development of this disease.

## 3. Materials and Methods

### 3.1. Yeast Strains and Growth Conditions

*S. cerevisiae* strains used in this study are listed in Table 1. Yeast cells were grown aerobically at 26 °C in an orbital shaker (at 140 rpm), with a ratio of flask volume:medium volume of 5:1. The growth medium used was synthetic complete (SC) medium, containing drop-out, 2% (*w*/*v*) glucose (ThermoFisher Scientific, Waltham, MA, USA), and 0.67% (*w*/*v*) yeast nitrogen base without amino acids (BD BioSciences, San Jose, CA, USA), supplemented with appropriate amino acids or nucleotides [0.008% (*w*/*v*) histidine (Sigma Aldrich, St. Louis, MO, USA), 0.038% (*w*/*v*) methionine (Sigma Aldrich, St. Louis, MO, USA), 0.04% (*w*/*v*) leucine (Sigma Aldrich, St. Louis, MO, USA), and 0.008% (*w*/*v*) uracil (Sigma Aldrich, St. Louis, MO, USA)]. Deletion of *AFT1* in *ncr1*Δ cells was performed using a deletion fragment containing *HIS3* and the flanking regions of *AFT1*. Deletion of *YCK3* in wild-type cells was performed using a deletion fragment containing *KanMX4* and the flanking regions of *YCK3*. Yeast cells were transformed using the lithium acetate/single-stranded carrier DNA/PEG method as described [96]. Cells were selected by growing in a medium lacking histidine or containing geneticin (300 µg mL^−1^), and gene deletion was confirmed by standard PCR procedures.

### 3.2. Vacuolar Membranes Isolation

#### 3.2.1. Spheroplasts Preparation and Lysis

Membrane vacuoles were isolated as previously described [97]. Cells were grown in 1 L of SC medium until the late exponential phase (OD_600 nm_ ≅ 2) and harvested by centrifugation at 5000 rpm for 8 min (rotor JLA-8.1000; high-speed centrifuge Avanti J-26 XP, Beckman, Brea, CA, USA). Cells were washed with 50 mL of double-distilled water and centrifuged again. The wet weight of cells from a 1 L culture was 2–3 g. Cells were resuspended in 2.5 mL of 1.1 M sorbitol (Merck, Darmstadt, Germany) plus 4.5 mg zymolyase (AMSBIO, Cambridge, MA, USA) and 2.5 µL of β-mercaptoethanol (Merck, Darmstadt, Germany) per 1 g of cells and incubated at 37 °C with shaking at 80 rpm. Light microscopy was used to evaluate the extent of spheroplast formation. After digestion of the cell wall, the spheroplasts were cooled on ice. Each 2.5 mL of spheroplast suspension was pipetted on top of a layer of 2.5 mL of ice-cold solution containing 7.5% (*w*/*v*) Ficoll (Corning, NY, USA) and 1.1 M sorbitol in a centrifuge tube. The spheroplasts were washed through this Ficoll-sorbitol layer by centrifugation at 5000 rpm for 20 min at 4 °C. The spheroplasts were lysed by a 6-fold dilution of the spheroplast pellet with the ice-cold lysis buffer containing 10 mM Tris-MES, pH 6.9, 12% (*w*/*v*) Ficoll, 0.1 mM MgCl_2_. After this step, all solutions were supplemented with a cocktail of protease inhibitors (Sigma Aldrich, St. Louis, MO, USA) and phosphatase inhibitors [1 mM sodium orthovanadate (Sigma Aldrich, St. Louis, MO, USA), 50 mM sodium fluoride (Sigma Aldrich, St. Louis, MO, USA), 5 mM sodium pyrophosphate (Fluka, Buchs, Switzerland)]. The suspension was homogenized on ice in a Dounce homogenizer (40 mL) by 15 strokes with a large clearance pestle “A” (Kontes Glassware, Vineland, NJ, USA). The cells and solutions were kept at 4 °C throughout the procedure.

#### 3.2.2. Isolation of Intact Vacuoles

Samples of the spheroplast lysate were transferred to a centrifuge tube and overlaid with lysis buffer until 38 mL. Centrifugation was performed in a swing-out bucket rotor (SW32Ti, Beckman, Brea, CA, USA) at 20,000 rpm for 30 min at 4 °C. The fraction floating on top of the tube, containing the crude vacuoles, was collected and resuspended in lysis buffer by homogenization with a loosely fitting Dounce homogenizer (5–6 strokes, pestle A). The homogenized crude vacuoles were overlaid in a centrifugation tube with a layer of 14 mL of 10 mM Tris-MES pH 6.9, 8% (*w*/*v*) Ficoll and 0.5 mM MgCl_2_ plus a second layer of 14 mL of the same buffer containing 4% (*w*/*v*) Ficoll. After centrifugation at 20,000 rpm for 45 min, intact vacuoles were floating on top of the 4% (*w*/*v*) Ficoll solution as a white wafer. Purified vacuoles were collected with a spoon-shaped spatula prewetted in 4% (*w*/*v*) Ficoll buffer.

#### 3.2.3. Isolation of Vacuolar Membranes

Vacuoles were lysed osmotically in the same volume of buffer consisting of 20 mM triethylammonium bicarbonate (TEAB) pH 8.0, 10 mM MgCl_2,_ and 50 mM KCl. The lysates were then diluted with 2 volumes of buffer containing 10 mM TEAB pH 8.0, 5 mM MgCl_2,_ and 25 mM KCl. Immediately (without incubation), vacuolar membranes were recovered by centrifugation at 80,000 rpm for 1 h (rotor TLA100.3, Beckman, Brea, CA, USA). To remove peripheral proteins, the pellet was resuspended in 100 mM sodium carbonate pH 11.8, and subsequently incubated in the same buffer plus 2 mM EDTA for 15 min on ice. The membranes were recovered by centrifugation for 3 h at 80,000 rpm (fixed angle rotor 70 Ti, Beckman, Brea, CA, USA). The protein concentration was determined using the Pierce BCA Protein Assay Kit (ThermoFisher Scientific, Waltham, MA, USA), after solubilization in 2% (*w*/*v*) SDS.

### 3.3. Proteomics

#### 3.3.1. Sample Preparation and Data Acquisition

Proteins were solubilized with 100 mM Tris pH 8.5, 1% (*w*/*v*) sodium deoxycholate, 10 mM tris(2-carboxyethyl)phosphine (TCEP), and 40 mM chloroacetamide and incubated for 10 min at 95 °C at 1000 rpm (Thermomixer, Eppendorf, Hamburg, Germany). Each sample was processed for proteomics analysis following the solid-phase-enhanced sample-preparation (SP3) protocol as described [98]. Enzymatic digestion was performed by adding Trypsin/LysC (2 μg) and incubating overnight at 37 °C at 1000 rpm. Protein identification and quantitation were performed by nanoLC-MS/MS as previously described [99], using an Ultimate 3000 liquid chromatography system coupled to a Q-Exactive Hybrid Quadrupole-Orbitrap mass spectrometer (ThermoFisher Scientific, Waltham, MA, USA).

#### 3.3.2. Data Analysis

The raw data were processed using Proteome Discoverer 2.5.0.400 software (ThermoFisher Scientific, Waltham, MA, USA) and searched against the UniProt database for the *S. cerevisiae* Proteome 2020_03 (6049 entries) together with a common contaminant database from MaxQuant (version 1.6.2.6, Max Planck Institute of Biochemistry, Munich, Germany). The Sequest HT search engine was used to identify tryptic peptides. The ion mass tolerance was 10 ppm for precursor ions and 0.02 Da for fragment ions. The maximum allowed missing cleavage sites was set to 2. Cysteine carbamidomethylation was defined as a constant modification. Methionine oxidation, asparagine and glutamine deamidation, serine, threonine and tyrosine phosphorylation, peptide N-terminus pyro-glutamine, protein N-terminus acetylation, Met-loss, and Met-loss + acetylation, were defined as variable modifications. Peptide confidence was set to high. The processing node Percolator was enabled with the following settings: maximum delta Cn 0.05; decoy database search target FDR 1%, validation based on q-value. The ptmRS node was used for the localization of modification sites. Protein label-free quantitation was performed with the Minora feature detector node at the processing step. Precursor ions quantification was performed at the processing step with the following parameters: peptides to use unique plus razor, precursor abundance based on intensity, and normalization based on the total peptide amount of annotated vacuolar membrane proteins. The percentage of phosphorylation for each residue was manually calculated by the ratio between the amount of phosphorylated peptide and the total peptide amount.

To identify group differences concerning protein levels or the percentage of phosphorylation, first, the homogeneity of variances was assessed with the Levene test. Afterwards, pairwise comparisons with the Student *t*-test were performed. Statistical analysis was performed using the Statistical Package for the Social Sciences (SPSS), version 26. The level of significance was set at α = 0.05. Statistical analysis of the overrepresentation of functional groups was performed by using the Gene Ontology database (1 July 2022 and 10.5281/zenodo.6799722).

### 3.4. Analysis of the AP-3 Pathway

Cells expressing pGNS416-*GFP-NYV1-SNC1* (a gift from Christian Ungermann, Osnabrück University, Germany) [60] were grown to exponential phase in SC medium and treated with 0.024 mM FM4-64 (Molecular Probes, Eugene, OR, USA) for 1 h. After washing, cells were observed by fluorescence microscopy (AxioImager Z1, Carl Zeiss, Oberkochen, Germany). Output final images were obtained using ImageJ 1.45v software [100].

### 3.5. Western Blotting

Yeast cells were grown to exponential or PDS phase in SC medium and collected by centrifugation. As a positive control of TORC1-mediated phosphorylation of ribosomal protein S6 (Rps6 in yeast) at S232/233 (S235/236 in human S6) and carboxypeptidase Y (CPY) maturation, wild-type cells grown to exponential phase were treated with rapamycin (200 ng mL^−1^) for 4 h. For protein extraction, cells were incubated in 0.1 M NaOH for 5 min at room temperature and harvested by centrifugation at 13,000 rpm (4 °C) for 15 min. The pellet was suspended in modified Laemmli buffer [62.5 mM Tris-HCl pH 6.8, 2% (*w*/*v*) SDS, 10% (*v*/*v*) glycerol, 0.002% (*w*/*v*) bromophenol blue], and incubated for 5 min at 95 °C. After centrifugation at 13,000 rpm (4 °C) for 3 min, the supernatant was collected, and the protein content was estimated by Pierce BCA Protein Assay Kit (ThermoFisher Scientific, Waltham, MA, USA), using bovine serum albumin (Sigma Aldrich, St. Louis, MO, USA) as a standard. To analyze CPY and ApeI levels or Rps6 phosphorylation at S232/233, protein samples (20 μg) were mixed with 5% (*v*/*v*) 2-mercaptoethanol before loading, separated by SDS-PAGE using 10% polyacrylamide gels (Grisp, Porto, Portugal), and blotted onto a nitrocellulose membrane (GE Healthcare, Chicago, IL, USA). Immunodetection was performed using anti-CPY (1:3000; Invitrogen, Waltham, MA, USA), anti-ApeI (1:200; Santa Cruz Biotechnology, Dallas, TX, USA), or anti-phospho-Rps6 (S235/236) (1:5000; Cell Signaling, Danvers, MA, USA) as primary antibodies, and anti-rabbit IgG-peroxidase (1:5000; Sigma Aldrich, St. Louis, MO, USA) or anti-goat IgG-peroxidase (1:5000; Sigma Aldrich, St. Louis, MO, USA) as secondary antibodies. To analyze the autophagic flux, cells expressing pRS416-*GFP-ATG8* were grown to exponential and PDS phases in SC medium or SC medium supplemented with 80 µM of BPS (disodium salt hydrate; Sigma Aldrich, St. Louis, MO, USA). As a positive control, wild-type cells grown to exponential phase were treated with rapamycin (200 ng mL^−1^) for 4 h. Protein samples (20 μg) were mixed with 5% (*v*/*v*) 2-mercaptoethanol before loading, separated by SDS-PAGE using 12.5% polyacrylamide gels, and blotted onto a nitrocellulose membrane. The membranes were incubated with the primary antibody anti-GFP (1:5000; Roche, Basel, Switzerland) and with the secondary antibody anti-mouse IgG-peroxidase (1:5000; Molecular Probes, Eugene, OR, USA). As loading control, Pgk1 levels were analyzed using anti-Pgk1 (1:30,000; Invitrogen, Waltham, MA, USA) as the primary antibody and anti-mouse IgG-peroxidase (1:10,000; Molecular Probes, Eugene, OR, USA) as the secondary antibody. The TORC1-dependent C-terminal phosphorylation of Sch9 was analyzed by Western blotting, using cells transformed with pRS416-*SCH9-5HA*. NTCB (2-nitro-5-thiocyanatobenzoic acid)-chemical fragmentation analysis [101] and immunodetection [102] were performed as described. Immunoblot was revealed by chemiluminescence (ECL, Advansta, San Jose, CA, USA).

### 3.6. Iron Levels

Yeast cells (6 × 10^8^ cells mL^−1^) were grown in SC medium to exponential and/or PDS phases, and total iron levels were quantified using a colorimetric assay, as previously described [75].

### 3.7. β-Galactosidase Assay

Wild-type and *ncr1*Δ cells were transformed with p*CTH2-LacZ*, a β-galactosidase reporter construct containing the consensus Aft1 binding sequences from *CTH2* promoter (a gift from Dennis Thiele, Duke University Medical Center, Durham, NC, USA) [103]. Cells were grown to exponential phase in SC medium, treated or not with 20 μM of BPS for 4 h, and β-galactosidase activity was assayed as previously described [75], using 160 μg of total protein or 20 μg of total protein for samples treated with BPS.

### 3.8. Chronological Lifespan

Chronological lifespan (CLS) was assessed as previously described [104]. Briefly, overnight cultures in SC medium were diluted to an OD_600 nm_ = 0.2 in SC or SC + 80 µM BPS. After 24 h (considered time zero, t0), cells were kept in culture medium at 26 °C, and cell viability at indicated days was determined by standard dilution plate counts on YPD medium containing 1.5% (*w*/*v*) agar. Colonies were counted after 2–3 days of incubation at 26 °C. Cellular viability was expressed as the percentage of the colony-forming units (CFUs) in relation to t0.

### 3.9. Oxidative Stress Resistance

For oxidative stress resistance analysis, cells were grown to PDS phase in SC medium supplemented or not with 80 µM BPS and exposed to 5 mM H_2_O_2_ (Merck, Darmstadt, Germany) for 1 h. Cell viability was determined as described above and expressed as the percentage of the CFUs (treated cells vs. non-stressed cells).

### 3.10. Intracellular Oxidation

Cells were grown to the PDS (48 after log) phase in SC medium supplemented or not with 80 µM BPS and the levels of intracellular ROS were detected with dihydrorhodamine (DHR) 123 (Molecular Probes, Eugene, OR, USA) by flow cytometry. A total of 3 × 10^7^ cells were treated with 6 μL of DHR 123 (stock solution at 2.5 mg mL^−1^; prepared in DMSO) and incubated for 30 min at 26 °C in the dark. Cells were washed twice with phosphate-buffered saline (PBS) solution and resuspended in PBS. The fluorescence of DHR-positive cells was measured for 80,000 events on the FL-1 channel (533/30), using the BD Accuri C6 Flow cytometer and the FlowJow software (v 10.8.1).

### 3.11. Oxygen Consumption

Cells were grown to the PDS phase in SC medium, and the oxygen consumption rate was measured for 1 × 10^7^ cells in PBS, using a Clark-type oxygen electrode coupled to an Oxygraph plus system (Hansatech, Norfolk, UK). Data were analyzed using the OxyTrace + software v1.0.48.

### 3.12. Statistical Analysis

Data were analyzed in GraphPad Prism Software v9.4.1. Values were compared by one-way ANOVA, two-way ANOVA, or by Student’s *t*-test * *p* ≤ 0.05; ** *p* ≤ 0.01; *** *p* ≤ 0.001; **** *p* ≤ 0.0001.

## Figures and Tables

**Figure 1 ijms-24-06221-f001:**
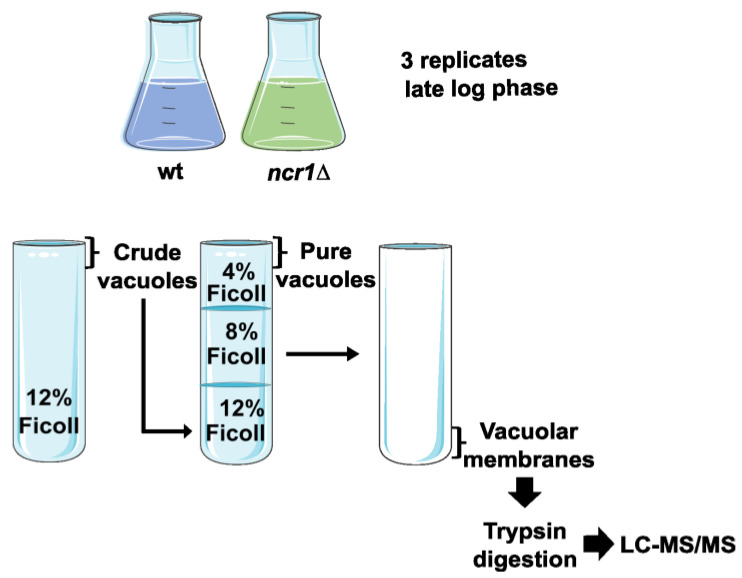
Overview of the experimental design. Wild-type (wt) and *ncr1*∆ cells were grown in SC medium to late exponential phase [late log; (OD_600 nm_ ≅ 2)]. Vacuoles were isolated by density centrifugations using Ficoll gradients and vacuolar membranes recovered after osmotic lysis of vacuoles. Samples were then digested with trypsin, and protein identification and quantitation were performed by nanoLC-MS/MS.

**Figure 2 ijms-24-06221-f002:**
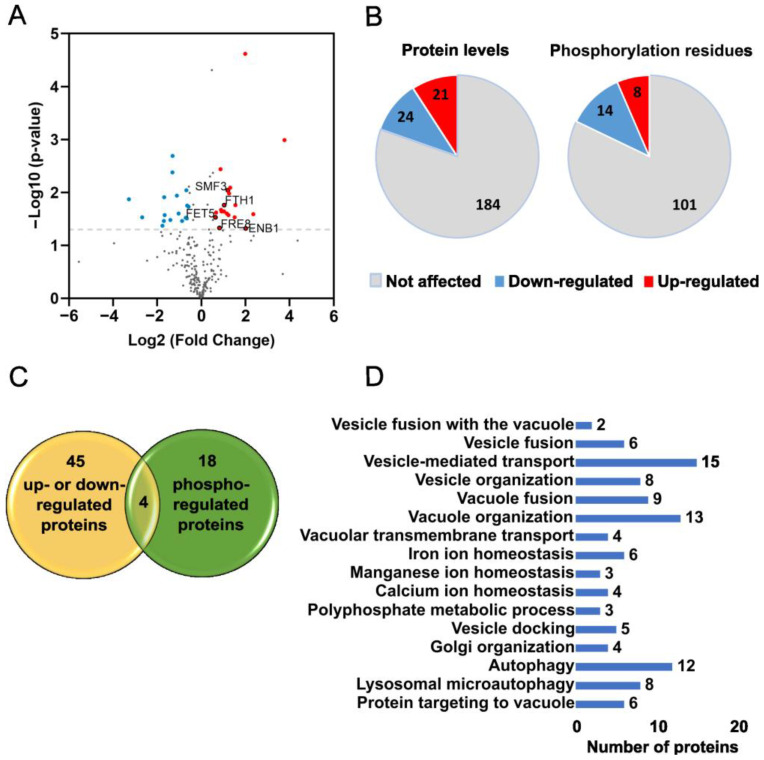
Overview of the phosphoproteomic analysis from *ncr1*∆ vacuolar membranes. (**A**) Volcano plot for differentially expressed proteins in *ncr1*∆ vs. wild-type samples. Log-transformed *p*-values (Student’s *t*-test) in the vertical axis are plotted against log-transformed fold change in the horizontal axis. The horizontal dashed line marks a *p*-value of 0.05. Red plots represent the up-regulated proteins with a 1.5-fold change threshold, and blue plots represent the down-regulated proteins (log2 fold change of +/−0.58). Red plots with black outlines represent the proteins associated with iron homeostasis. (**B**) Number of proteins and number of phosphorylation sites significantly up- or down-regulated in the *ncr1*∆ mutant. (**C**) Overlap of proteins with alterations in the levels and phosphorylation in *ncr1*∆ samples. (**D**) Gene Ontology (GO) term enrichment analysis on biological processes was performed for proteins with statistical alterations in *ncr1*∆ samples (levels and phosphorylation).

**Figure 3 ijms-24-06221-f003:**
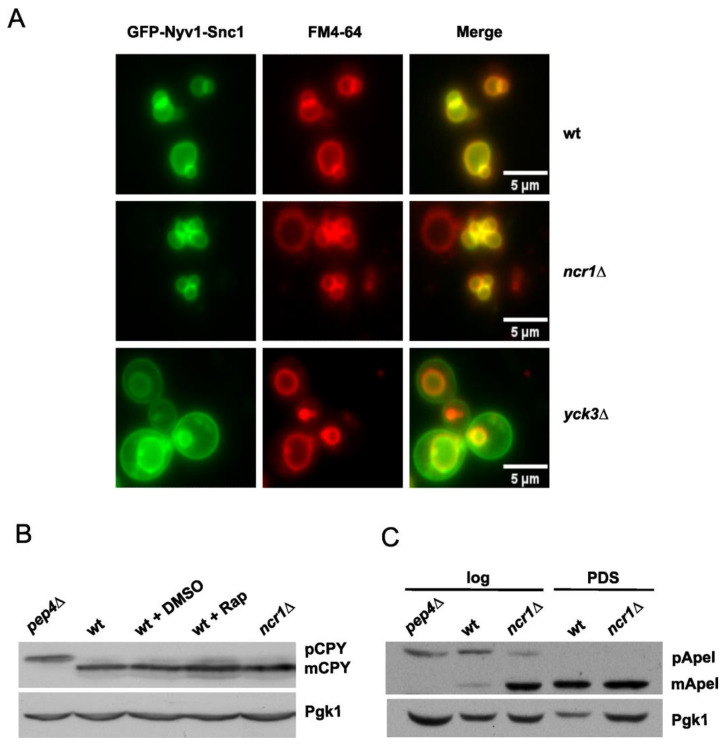
Loss of Ncr1 does not impair protein trafficking to the yeast vacuole. (**A**) ALP-AP-3 pathway: wild-type (wt), *ncr1*∆, and *yck3*∆ cells carrying a plasmid expressing a GFP-tagged Nyv1-Snc1 fusion protein grown to exponential phase were incubated with 0.024 mM FM4-64 (vacuolar membrane probe) for 1 h and observed by fluorescence microscopy. (**B**) CPY pathway: cells were grown to exponential phase and proteins were analyzed by immunoblotting, using anti-CPY antibody. Bands corresponding to the proenzyme (pCPY) or the mature enzyme (mCPY) are indicated. Pgk1 was used as loading control. Cells lacking Pep4, which is involved in the proteolytic activation of pCPY to mCPY, were used as a negative control. As a positive control, wild-type cells were treated with rapamycin (200 ng mL^−1^) for 4 h (wt + Rap). The original figure is shown in Appendix A. (**C**) Cvt pathway: cells were grown to exponential (log) and post-diauxic shift [(PDS), 48 h after log] phases, and the processing of pApeI to mApeI was analyzed by immunoblotting, using anti-ApeI antibody. Pgk1 was used as loading control. Cells lacking Pep4, which is involved in the proteolytic activation of pApeI to mApeI, were used as control. The original figure is shown in Appendix A. Representative images of at least three independent experiments are shown.

**Figure 4 ijms-24-06221-f004:**
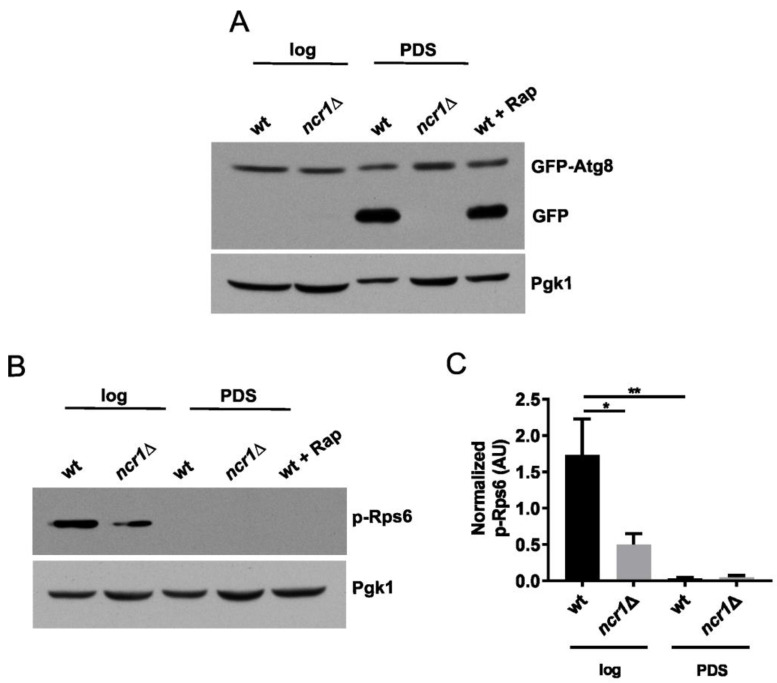
Cells lacking Ncr1 exhibit autophagic defects despite reduced TORC1 activity. (**A**) Wild-type (wt) and *ncr1*∆ cells carrying pRS416-*GFP-ATG8* were grown to the exponential (log) and post-diauxic shift [(PDS), 48 h after log] phases, and proteins analyzed by immunoblotting using anti-GFP antibody. As positive control, wild-type cells grown to exponential phase were treated with rapamycin (200 ng mL^−1^) for 4 h (wt + Rap). The original figure is shown in Appendix A. (**B**) Cells were grown to exponential (log) and post-diauxic shift [(PDS), 48 h after log] phases, and the levels of phospho-Rps6 (S232/233), used as read-out for TORC1 activity, were evaluated by Western blotting. Pgk1 was used as loading control. Representative images of at least three independent experiments are shown. The original blots are shown in Appendix A. (**C**) Quantification of phospho-Rps6 levels were calculated by the ratio between p-Rps6 and Pgk1 signals. Values are the mean ± SEM. * *p* ≤ 0.05; **, *p* ≤ 0.01 (one-way ANOVA).

**Figure 5 ijms-24-06221-f005:**
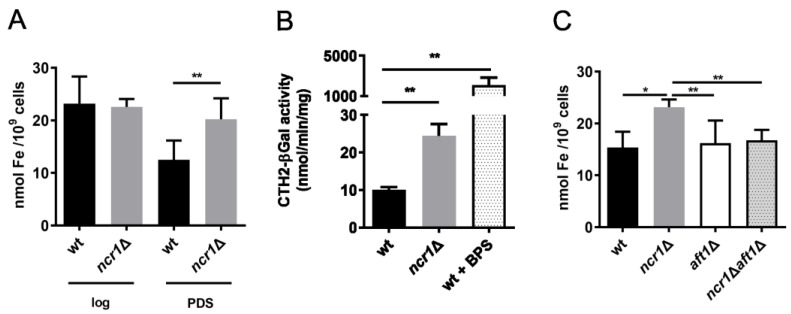
Ncr1-deficient cells accumulate iron at post-diauxic shift (PDS) phase in an Aft1-dependent manner. (**A**) Total iron levels were quantified in wild-type (wt) and *ncr1*∆ cells grown to exponential (log) and PDS (48 h after log) phases. (**B**) Wild-type and *ncr1*Δ cells expressing the Aft1 binding sequence from *CTH2* promoter fused to a LacZ reporter were grown to the exponential phase, and β-galactosidase activity was determined. Wild-type cells treated with 20 µM of bathophenanthrolinedisulfonic acid (BPS) for 4 h (wt + BPS) were used as positive control. (**C**) Iron levels were quantified in wt, *ncr1*Δ, *aft1*Δ, and *ncr1*Δ*aft1*Δ cells grown to the PDS (48 h after log) phase. Data are the mean ± SD of at least three independent experiments. * *p* ≤ 0.05; ** *p* ≤ 0.01 (Student’s *t*-test).

**Figure 6 ijms-24-06221-f006:**
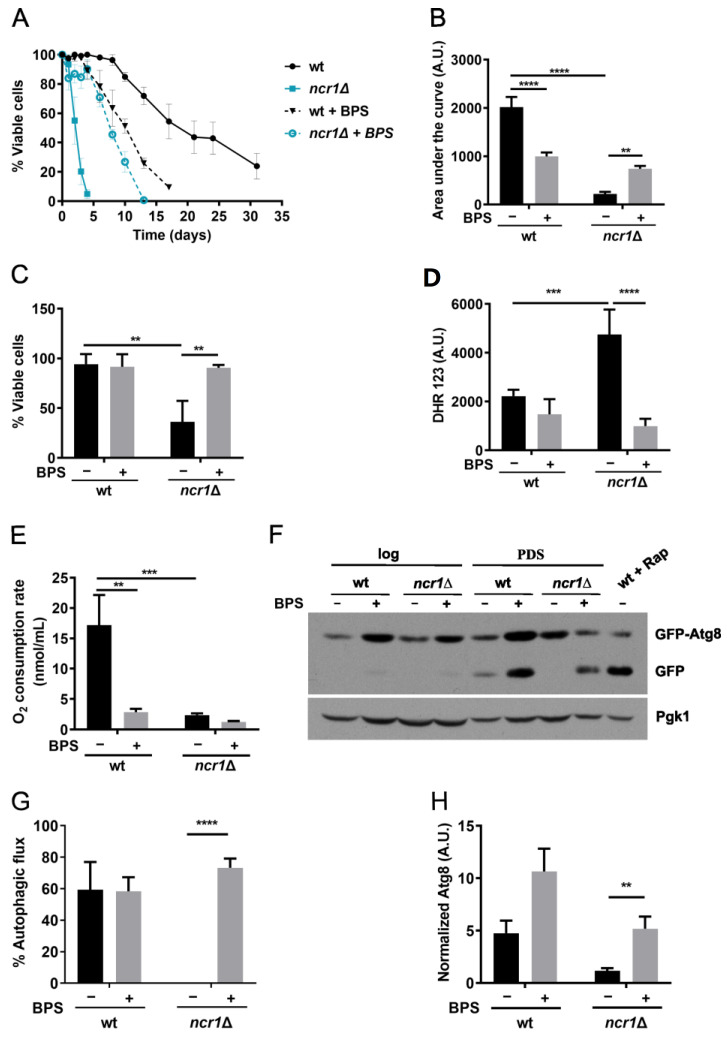
Chelating iron with BPS increases the chronological lifespan, oxidative stress resistance, and autophagic flux of *ncr1*∆ cells. (**A**) Wild-type and *ncr1*Δ cells were grown to post-diauxic shift [(PDS), 24 h after log] phase and maintained overtime in SC medium (wt, *ncr1*∆) or SC medium supplemented with 80 µM of BPS (wt + BPS, *ncr1*∆ + BPS). Cellular viability was expressed as the percentage of the colony-forming units in relation to day 0. Data are the mean ± SEM of at least three independent experiments. (**B**) The area under each lifespan curve was computed using GraphPad in arbitrary units (A.U.). Data are the mean ± SEM. ** *p* ≤ 0.01; ****, *p* ≤ 0.0001 (one-way ANOVA). (**C**) Cells were grown to PDS (24 h after log) phase in medium supplemented or not with 80 µM of BPS and exposed to 5 mM H_2_O_2_ for 1 h. Cell viability was measured as the percentage of the colony-forming unit (treated cells vs. non-stressed cells). Values are the mean ± SD. ** *p* ≤ 0.01 (two-way ANOVA). (**D**) Cells were grown to PDS (48 h after log) phase in medium supplemented or not with 80 µM of BPS and intracellular reactive oxygen species were analyzed by flow cytometry, using Dihydrorhodamine 123 (DHR123) as probe. Data are the mean ± SD. *** *p* ≤ 0.001; **** *p* ≤ 0.0001 (two-way ANOVA). (**E**) Cells were grown to PDS (24 h after log) phase in medium supplemented or not with 80 µM of BPS and oxygen consumption rates were measured. Values are the mean ± SD (two-way ANOVA), ** *p* ≤ 0.01, *** *p* ≤ 0.001. (**F**) Cells carrying pRS416*-GFP-ATG8* were grown to exponential (log) and PDS (48 h after log) phases in medium supplemented or not with 80 µM of BPS. Proteins were analyzed by immunoblotting using anti-GFP antibody. As positive control, wild-type cells grown to exponential were treated with rapamycin (200 ng mL^−1^) for 4 h (wt + Rap). A representative blot of at least three independent experiments is shown. The original blots are shown in Appendix A. (**G**) Autophagic flux was calculated by the ratio between the free GFP signal and the sum of free GFP and GFP-Atg8 signals. Values are the mean ± SEM. **** *p* ≤ 0.0001 (Multiple Student’s *t*-tests). (**H**) Atg8 levels were calculated by the ratio between free GFP plus GFP-Atg8 and Pgk1 signals. Values are the mean ± SEM. ** *p* ≤ 0.01 (Multiple Student’s *t*-tests).

**Figure 7 ijms-24-06221-f007:**
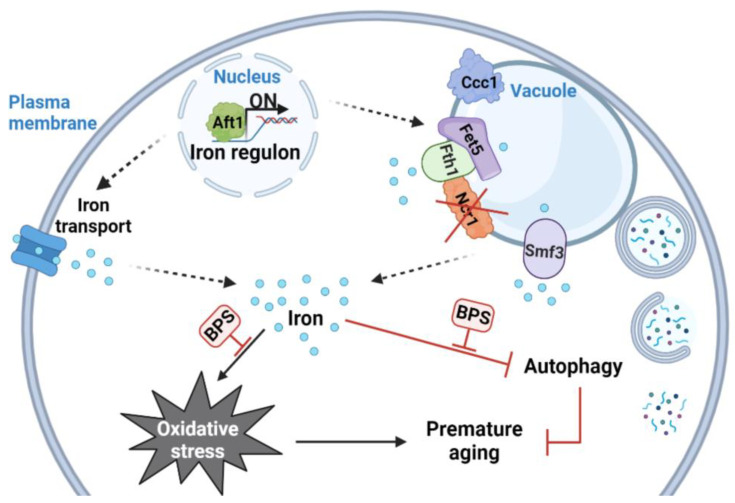
Working model. Absence of Ncr1 leads to Aft1 activation, increasing the transcription of the iron regulon and causing iron dyshomeostasis. Our results show that the levels of vacuolar iron exporters (Fet5, Fth1, and Smf3) increase in vacuolar membranes of *ncr1*∆ cells. The expression of plasma membrane iron transporters may also be induced (due to Aft1 activation), but this remains to be demonstrated. Iron accumulation promotes oxidative stress and blocks autophagy. Iron limitation, by treating cells with BPS, decreases oxidative stress, restores the autophagic flux, and, consequently, improves the lifespan in the yeast model of NPC1. Dashed arrows represent cellular functions and mechanisms demonstrated in other studies.

**Table 1 ijms-24-06221-t001:** *Saccharomyces cerevisiae* strains used in this study.

Strain	Genotype	Source
BY4741 ^a,b,c,d^	Mata, *his3*Δ1, *leu2*Δ0, *met15*Δ0, *ura3*Δ0	EUROSCARF
*ncr1*Δ ^a,b,c,d^	BY4741 *ncr1*::*KanMX4*	[56]
*yck3*∆ ^a^	BY4741 *yck3*::*KanMX4*	This study
*pep4*∆	BY4741 *pep4*::*KanMX4*	EUROSCARF
*aft1*Δ	BY4741 *aft1*::*HIS3*	[75]
*ncr1*Δ*aft1*Δ	BY4741 *ncr1*::*KanMX4 aft1*::*HIS3*	This study

Cells harboring ^a^ pGNS416-*GFP-NYV1-SNC1*, ^b^ pRS416-*GFP-ATG8,* ^c^ p*CTH2*-LacZ, and ^d^ pRS416-*SCH9*-*5HA* are indicated.

## Data Availability

The mass spectrometry proteomics data have been deposited to the ProteomeXchange Consortium via the PRIDE [105] partner repository with the dataset identifier PXD039881.

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
