# Peer review of "Iron Limitation Restores Autophagy and Increases Lifespan in the Yeast Model of Niemann–Pick Type C1"

_ijms, 2023, doi:10.3390/ijms24076221_

Round 1

Reviewer 1 Report

The authors present a very nice study in which characterize novel functions y targets for the gene by using budding yeast as model system. Given the high degree of homology in structure and function of the gene and the important consequences that the correct function of this gene has in human health this paper is relevant and very useful in human research. However, there are several aspects that should be addressed previous publication:

Figure 3C requires explanation in the context of this study

The authors might also explain the biological relevance of exploring PSD versus stationary phase and the biological interpretation of exponential phase results which are totally different to those found in PDS. Does PDS have any specific relevance as model to study a human disease?

Authors claim that Torc1 is downregulated in npc1 mutant, however, rapamycin treatment restores autophagy in the mutant, meaning that TORC1 is inhibited for some substrates, probably nor for all of them, authors should show more readouts in order to stablish the magnitude of TORC1 inhibition before coming to big conclusions

Fig 4b suggest that Rps8 phosphorylation is regulated by Npc1 independently of TORC1, however this statement is not completely demonstrated, there exists the possibility that Npc1 regulates Rps8 phosphorylation and /or autophagy flux independently on TORC1. The authors should address this point with additional experimental data in order to sustain their hypothesis.

Fig 5a needs an accompanying growth curve since it is necessary to sort out whether cell growth rate could be delaying the entrance into postdiauxia in the mutant. Authors should indicate whether iron accumulation is restricted to PDS or whether it is also detected during stationary phase. In addition, authors should show whether this iron accumulation is causing oxidative stress (ethidium bromide staining is a fast method to check this, at least qualitatively) and if this is the case, it would be necessary to ascertain whether the phenotypic effects shown in the mutant are due to oxidative stress.

Images or quantification. It would be nice to know where iron is accumulated in the mutant.

CLS experiments are no coincidental with similar experiments recently published. In this paper they are performed differently than standards. Colony forming colonies should be compared to values obtained upon at least 3 days of growth when cells are supposed to be into stationary phase. Authors should follow:

Postnikoff SD, Harkness TA. Replicative and chronological life-span assays. Methods Mol Biol. 2014;1163:223-7. doi: 10.1007/978-1-4939-0799-1_17. PMID: 24841311.

Or

Fabrizio P, Longo VD. The chronological life span of Saccharomyces cerevisiae. Methods Mol Biol. 2007;371:89-95. doi: 10.1007/978-1-59745-361-5_8. PMID: 17634576.

The figure 7 should be revised along with the explanatory legend since the arrows draw mechanisms not demonstrated in this study, in that case arrows should be discontinuous and clearly commented in the text. In this respect, Ccc1 phosphorylation in the figure is not sustained for experimental results obtained in this paper and related to Npc1, this should be clarified.

Author Response

Figure 3C requires explanation in the context of this study

R: Our proteomic studies showed a 13.6-fold increase of Ams1 in the vacuolar membranes of ncr1∆ mutant cells. This is one of the hydrolases that is transported to vacuoles through the cytoplasm to vacuole targeting (Cvt) pathway, which led us to assess the activation of Cvt. For that, we analyzed the processing of ApeI to its mature form (Figure 3C), since this is a well-established assay for Cvt. This was highlighted by changing the text to: “Our proteomic analysis showed a 13.6-fold increase of Ams1 in the vacuolar membranes of ncr1∆ mutant. This led us to postulate that the Cvt pathway may be induced in this mutant. To test this hypothesis, we analyzed the processing of ApeI to its mature form by Western blotting, a well-established assay for Cvt. Since this pathway is induced during growth from exponential (fermentative) to post-diauxic shift (PDS; respiratory) phase, we assessed Cvt in both phases. The results demonstrated that the levels of the mature form of ApeI increased in ncr1∆ cells grown to the exponential phase (Figure 3C), suggesting that the Cvt pathway flux is enhanced in cells lacking Ncr1. It should be noted that ApeI was not detected in our study since it is not a vacuolar membrane protein.”.

The authors might also explain the biological relevance of exploring PSD versus stationary phase and the biological interpretation of exponential phase results which are totally different to those found in PDS. Does PDS have any specific relevance as model to study a human disease?

R: Some pathways, such Cvt and autophagy, are induced during growth to the PDS/respiratory and stationary phases. These cellular adaptations are important for cell longevity and its impairment has been associated with numerous diseases, including neurodegenerative disorders, and aging. To clarify these aspects, we have made some changes to the text.

Authors claim that Torc1 is downregulated in npc1 mutant, however, rapamycin treatment restores autophagy in the mutant, meaning that TORC1 is inhibited for some substrates, probably nor for all of them, authors should show more readouts in order to stablish the magnitude of TORC1 inhibition before coming to big conclusions

R: Rapamycin induced autophagy in wild type cells (used as control) but we did not assess its effect in ncr1∆ cells. Taking into account the reviewer's suggestion, we performed another assay, using Sch9 phosphorylation as readout. These results showed that Sch9 phosphorylation also decreases in ncr1∆ cells (Supplementary Figure S1), supporting our hypothesis that TORC1 is inhibited in this mutant.

Fig 4b suggest that Rps8 phosphorylation is regulated by Npc1 independently of TORC1, however this statement is not completely demonstrated, there exists the possibility that Npc1 regulates Rps8 phosphorylation and /or autophagy flux independently on TORC1. The authors should address this point with additional experimental data in order to sustain their hypothesis.

R: We did not suggest that Rps6 phosphorylation is TORC1-independent. We used Rps6 phosphorylation as readout of TORC1 activity. Since it decreased in ncr1∆ mutant cells, we suggest that the impairment of autophagy is TORC1-independent. To confirm TORC1 inhibition in the ncr1∆ mutant, we also measured Sch9 phosphorylation (please see the response to the previous comment).

Fig 5a needs an accompanying growth curve since it is necessary to sort out whether cell growth rate could be delaying the entrance into postdiauxia in the mutant. Authors should indicate whether iron accumulation is restricted to PDS or whether it is also detected during stationary phase. In addition, authors should show whether this iron accumulation is causing oxidative stress (ethidium bromide staining is a fast method to check this, at least qualitatively) and if this is the case, it would be necessary to ascertain whether the phenotypic effects shown in the mutant are due to oxidative stress.

R: We included the growth curves of wt and ncr1∆ cells as Supplementary Figure S2. Although cell growth decreases in the mutant strain, the analysis at PDS phase was performed at time points in which both wt and ncr1∆ cells had clearly made the diauxic shift. Thus, it is unlikely that the differences in iron levels are due to a growth delay in ncr1∆ cells. We also assessed intracellular oxidation in cells untreated and treated with the iron chelator BPS. Our results show that BPS decreased the high ROS levels exhibited by ncr1∆ vs wt cells, supporting our hypothesis that iron accumulation is causing oxidative stress in the mutant strain. These results were included as Figure 6D and discussed in the text. Since BPS also increased chronological lifespan and oxidative stress resistance of ncr1∆ cells, overall we believe that the results show that oxidative stress contributes to ncr1∆ phenotypes.

Images or quantification. It would be nice to know where iron is accumulated in the mutant.

R: We agree that the characterization of where iron accumulates may be important to understand the mechanism of toxicity in NPC. Since proteins involved in vacuolar iron mobilization were upregulated and vacuolar iron storage is known to protect against iron toxicity, iron is probably accumulating in the cytoplasm and, as consequence, it may also accumulate in other organelles. We think this issue requires several studies and should be the focus of another work and publication.

CLS experiments are no coincidental with similar experiments recently published. In this paper they are performed differently than standards. Colony forming colonies should be compared to values obtained upon at least 3 days of growth when cells are supposed to be into stationary phase.

R: We understand the point raised by the referee. We also have several studies in which we allow cells to grow for 3 days to reach stationary phase (t0 of aging). However, the ncr1∆ cells die quickly, which prevent us from using those standard conditions used in most studies. Given that we compare the viability of these cells with that of wild type cells at the same time points, we believe that the assay (used in our previous studies, Vilaça et al., 2014 and 2018) and conclusions taken are valid.

The figure 7 should be revised along with the explanatory legend since the arrows draw mechanisms not demonstrated in this study, in that case arrows should be discontinuous and clearly commented in the text. In this respect, Ccc1 phosphorylation in the figure is not sustained for experimental results obtained in this paper and related to Npc1, this should be clarified.

R: We thank the reviewer and changed the figure and legend taking into account the suggestions. Regarding Ccc1 phosphorylation, we agree to remove it from the figure since its putative effect on iron transport is unknown and not assessed in this study. Figure legend was altered according to these alterations.

Reviewer 2 Report

The manuscript by Martins et al. describes a yeast model of NPC1 disease.  By performing phosphor proteomic analysis, they identified changes in protein level and phosphorylation in yeast lacking ncr1 (orthologue of human NPC1).  They showed that Cvt pathway is enhanced and autophagy is compromised in those cells.  Furthermore, they showed that correcting iron overload in ncr1D cells (by adding iron chelator BPS) restored its autophagy flux.  Overall, the manuscript is well-written, and the findings were informative.

However, while the findings were interesting, it fell short in connecting initial phosphoproteomics analysis and their further experiments.  In particular, the authors mentioned 4 proteins (Yck3, Ybt1, Fet5 and Fth1) presenting alterations in both levels and phosphorylation.  However, no discussion and no experiments were conducted to examine if the phosphorylation changes in these proteins affect AP3 pathway (for Yck3) or iron sensing and homeostasis (for Fet5, Fth1, and Ccc1).  It would be interesting to see if phosphorylation changes are involved in these pathways and contribute to ncr1D cell phenotypes.

Minor points:

Please specify the PDS growing conditions (how many hours) in the figure legends.

Author Response

While the findings were interesting, it fell short in connecting initial phosphoproteomics analysis and their further experiments.  In particular, the authors mentioned 4 proteins (Yck3, Ybt1, Fet5 and Fth1) presenting alterations in both levels and phosphorylation.  However, no discussion and no experiments were conducted to examine if the phosphorylation changes in these proteins affect AP3 pathway (for Yck3) or iron sensing and homeostasis (for Fet5, Fth1, and Ccc1).  It would be interesting to see if phosphorylation changes are involved in these pathways and contribute to ncr1D cell phenotypes.

R: Regarding the AP3 pathway, our findings suggest that it is not altered in ncr1∆ cells. Thus, it is unlikely that changes in Yck3 phosphorylation contributes to ncr1∆ phenotypes. This was discussed in the revised manuscript. Regarding iron transporters, we agree that it would be interesting and important to characterize how its dysregulation, including changes in its phosphorylation, impacts on ncr1∆ phenotypes, but we think these studies should be the focus of another work and publication. We also added a comment about this in the discussion.

Minor points:

Please specify the PDS growing conditions (how many hours) in the figure legends.

R: The information requested was added in figure legends.

Round 2

Reviewer 1 Report

Authors have addressed the majority of the points raised by the reviewer. However, there is one that is not convincingly addressed and contradicts previous published results. Authors use PBS in order to deplete for iron, and use a not conventional protocol to assess CLS. These factors could be contributing to contaminate somehow the results obtained in Fig 6A and to reach conclusions contradicting those published in Montella-Manuel S, Pujol-Carrion N, Mechoud MA, de la Torre-Ruiz MA. Bulk autophagy induction and life extension is achieved when iron is the only limited nutrient in Saccharomyces cerevisiae. Biochem J. 2021 Feb 26;478(4):811-837. doi: 10.1042/BCJ20200849. PMID: 33507238. In order to circumvent this problem authors should perform equivalent experimental approach which is to deplete iron in the culture media not using a quelator and to calculate CLS according to the standard procedures as indicated in the previous report. This point needs to be necessarily addressed since according to Montella et al. iron depletion (removing iron from the reagents used to build the culture medium meaning from the begining of the experiment) extends CLS in wt cells and according to the current manuscript iron depletion by using PBS (chelator) decreases CLS in wt cells.

Author Response

N/A

Reviewer 2 Report

N/A

Author Response

N/A